# Analysis of Fundamental Differences between Capacitive and Inductive Impedance Matching for Inductive Wireless Power Transfer

**Yelzhas Zhaksylyk** *[ID]**, Einar Halvorsen**[ID]**, Ulrik Hanke**[ID] **and Mehdi Azadmehr**

Department of Microsystems, University of South-Eastern Norway, Campus Vestfold, NO-3184 Horten, Norway;
Einar.Halvorsen@usn.no (E.H.); Ulrik.Hanke@usn.no (U.H.); Mehdi.Azadmehr@usn.no (M.A.)
* Correspondence: Yelzhas.Zhaksylyk@usn.no

**Abstract:** Inductive and capacitive impedance matching are two different techniques optimizing power transfer in magnetic resonance inductive wireless power transfer. Under ideal conditions, i.e., unrestricted parameter ranges and no loss, both approaches can provide the perfect match. Comparing these two techniques under non-ideal conditions, to explore fundamental differences in their performance, is a challenging task as the two techniques are fundamentally different in operation. In this paper, we accomplish such a comparison by determining matchable impedances achievable by these networks and visualizing them as regions of a Smith chart. The analysis is performed over realistic constraints on parameters of three different application cases both with and without loss accounted for. While the analysis confirms that it is possible to achieve unit power transfer efficiency with both approaches in the lossless case, we find that the impedance regions where this is possible, as visualized in the Smith chart, differ between the two approaches and between the applications. Furthermore, an analysis of the lossy case shows that the degradation of the power transfer efficiencies upon introduction of parasitic losses is similar for the two methods.

**Keywords:** impedance matching network; parasitic resistance; power loss; reflection coefficient; Smith chart; wireless power transfer

---

## 1. Introduction

Recent demand on mobility and accessibility of devices is pushing the development of wireless technology to new levels. There are good solutions for data transfer such as WiFi and Bluetooth, whereas, power is still delivered by either batteries or cable, the main bottleneck in the strive for cutting all the wires and limiting the mobility of devices.

For daily-life applications, inductive wireless power transmission has drawn increasing attention from researchers as it offers the highest power transfer efficiency (PTE) among other alternatives such as capacitive, microwave, laser, and acoustic [1,2]. Various products such as electric toothbrushes and mobile chargers using this technique are already commercially available. This technique provides two advantages compared to others: transfer of high power and low-frequency operation, making it less hazardous to the human body [3]. The main issue with this type of inductive wireless power transfer (WPT) is the mobility as the sender and receiver need to be close to each other, less than a few centimetres. Magnetic resonant (MR) WPT, an inductive technique based on highly coupled high-Q resonators, addresses this issue and offers a reasonable distance of power transfer (up to 2 m) [4]. However, a considerable challenge of MR-WPT is to maintain high power transfer throughout a range of distances between resonators and for variations in load value, as these will cause a mismatch between the source and input impedances [5].

In order to solve this challenge, different types of impedance matching techniques have been developed in the last decades. The simplest and most popular ones use capacitive or inductive impedance matching networks (IMNs). The capacitive method uses variable capacitors to tune the transmitter to the resonant frequency or a predefined capacitor sequence for different distances [6,7]. There is a variety of adaptive frequency tuning systems where $L$, $T$ and $\Pi$-type impedance matching networks contain capacitors [7,8]. Matching can potentially also provide power to multiple device WPT by using only a single transmitter [9].

The four-coil MR-WPT system presented by the MIT group in 2007 [4] has become a recognized solution [10–14] for highly resonant WPT systems for medium distances. The system consists of two or more high-Q resonating coils which are driven by a low-Q coil connected to the power source. The load is also connected to a low-Q coil. The coupling between resonator and the driver coils (or the load coil) can be considered as parts of a matching network, where tuning of the impedance can be achieved by changing the coupling between them. In our study, this method is referred to as inductive matching. In a previous work, we showed that these two matching techniques, i.e., the capacitive and inductive matching could potentially achieve a similar level of matching in certain cases [15]. Among the many different capacitive compensation circuitries, we chose the parallel-series compensation according to reference [8] for comparison to the inductive method. This network offers sufficient degrees of freedom to match perfectly if there is no loss and no restriction on parameter ranges, hence it is sufficient to give an insight into the effect of these limitations. The presented comparison method can also be used to identify matchable regions of other compensation structures.

The aforementioned matching techniques, i.e., capacitive and inductive can be applied to any mismatches in a WPT system to improve the power transfer efficiency [16]. However, there is a challenge in the direct comparison of their matching performances because they have different circuit topologies. The paper describes a method that makes a systematic comparison of their performance possible. The proposed method is based on comparison of the conjugate impedance of the matchable load, displayed in the Smith chart. The conjugate matching method has been analysed in [17], where all concepts of conservation and amplification of power by two port network was defined. This method is used by [18,19] to describe efficiency of the WPT system with inductive IMN. Our work presents a comparison of the matchable loads offered by the inductive and capacitive matching networks over a full range of realistic parameter ranges for three different applications distinguished by their operating frequency and power level [20–22]. The operating frequencies are within the allowed Industrial, Scientific and Medical (ISM) bands [9], which also limits the frequency range in the analysis. For the ease of simulation and calculations we choose to keep the distance constant and match the different load values. We assume the coil sizes are such that the systems operate in near field and the inductances of sender and receiver coils are equal, as in the [18–22].

This paper is organized as follows. In Section 2, impedances are analysed by derivation of reflection coefficients. We intentionally exclude the parasitic components in the system in order to have a clear comparison between these matching techniques in the ideal case. Subsequently, Section 3 visualizes the matchable reflection coefficients in the Smith chart, which graphically illustrates all of the possible complex impedances that are obtained by sweeping matching parameters. Therefore, it demonstrates which method offers the wider area of impedances that can be matched. Furthermore, the impact of parasitic loss to the matchable region is analyzed and optimized power simulation is given in the Section 4 and Section 5 discusses the outcome of the comparison.

## 2. Reflection Coefficients

In order to map and compare the tunable impedances of WPT systems, suitable circuit models and corresponding impedance expressions should be established. A generic WPT system with impedance matching networks can be represented as shown in Figure 1. The driving source consists of an ideal voltage source ($V_S$) and a series resistance $R_S$. The two-port network consists of resonators, and here

we will consider capacitive and inductive impedance matching circuits. The network is terminated by load impedance $Z_L$ at the output. Here, $i_S$ and $i_L$ are currents through $R_S$ and $Z_L$, respectively.

Matching networks are necessary to obtain a match between input impedance $Z_{in}$ and source impedance $R_S$. They affect the Power Transfer Efficiency (PTE), defined as ratio between power delivered to the load and input power of the two-port network [18]. This research focuses on the comparison of the matchable loads offered by the capacitive and inductive IMNs. Therefore, the two-port network is redrawn as in Figure 2 to get value of an effective impedance $Z_{out}$ at the output, which is a complex conjugate $Z_L^*$ form of load impedance that can be perfectly matched.

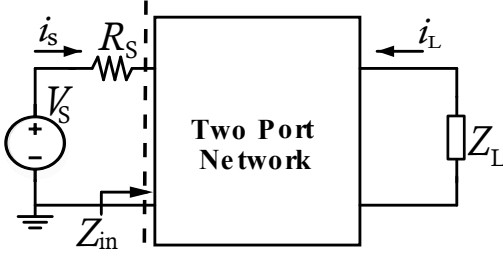

**Figure 1.** Two-port network representation of a highly resonant WPT system.

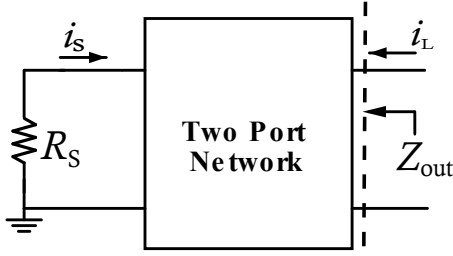

**Figure 2.** Two-port network representation when source is terminated.

### 2.1. Capacitive Matching Network

A lossless model of the inductive resonant WPT system with capacitive impedance matching is shown in Figure 3. The circuit elements are ideal, i.e., inductors and capacitors do not have parasitics. Matching networks consist of series-parallel connection of capacitors $C_{ts}$, $C_{tp}$ and $C_{rp}$, $C_{rs}$ at the transmitter (Tx) and receiver (Rx) sides. The source impedance is considered as resistance $R_S$.

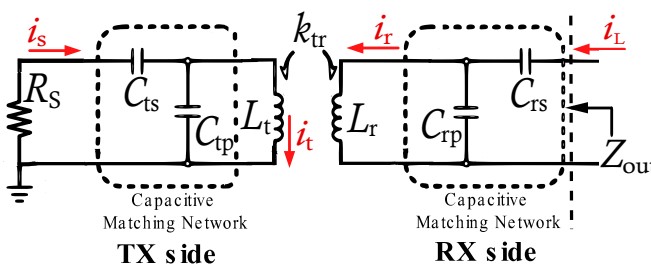

**Figure 3.** Equivalent circuit of a WPT system with capacitive impedance matching network.

Applying Kirchhoff's voltage law (KVL) to the circuit in Figure 3, the voltage-current relations can be written in an impedance matrix form

$$\begin{bmatrix} V_{out} \\ 0 \\ 0 \\ 0 \end{bmatrix} = \begin{bmatrix} Z_{11} & Z_{12} & 0 & 0 \\ Z_{21} & Z_{22} & Z_{23} & 0 \\ 0 & Z_{32} & Z_{33} & Z_{34} \\ 0 & 0 & Z_{43} & Z_{44} \end{bmatrix} \begin{bmatrix} i_L \\ i_r \\ i_t \\ i_S \end{bmatrix} \tag{1}$$

where

$$Z_{11} = \frac{1}{j\omega C_{rs}} + \frac{1}{j\omega C_{rp}}, \ Z_{12} = -\frac{1}{j\omega C_{rp}}, \tag{2}$$

$$Z_{21} = -\frac{1}{j\omega C_{rp}}, \ Z_{22} = j\omega L_r + \frac{1}{j\omega C_{rp}}, \ Z_{23} = j\omega M_{12}, \tag{3}$$

$$Z_{32} = j\omega M_{tr}, Z_{33} = j\omega L_t + \frac{1}{j\omega C_{tp}}, Z_{34} = -\frac{1}{j\omega C_{tp}}, \tag{4}$$

$$Z_{43} = -\frac{1}{j\omega C_{tp}}, \ Z_{44} = R_S + \frac{1}{j\omega C_{ts}} + \frac{1}{j\omega C_{tp}}. \tag{5}$$

Here,

$$M_{tr} = k_{tr}\sqrt{L_t L_r}, \ 0 \le k_{tr} \le 1 \tag{6}$$

is the mutual inductance between inductors $L_t$ and $L_r$. The coefficient $k_{tr}$ represents the coupling between them and its value is inversely proportional to the cube of their distance [15]. The distance change and variation of load impedance can be controlled by adjusting the capacitances $C_{ts}, C_{tp}$ and $C_{rs}, C_{rp}$ in the matching networks.

The effective impedance $Z_{out}$ at the output of two-port network is

$$Z_{out} = jX_r + \frac{Z_{12}^2(Z_{34}^2 - Z_{33}Z_{44})}{(Z_{22}Z_{33} - Z_{32}^2)Z_{44} - Z_{34}^2 Z_{22}}, \tag{7}$$

where

$$X_r = -\frac{1}{\omega C_{rs}} - \frac{1}{\omega C_{rp}}. \tag{8}$$

The real and imaginary parts of the impedance are

$$Re\{Z_{out}\} = \Delta R_S, \ Im\{Z_{out}\} = \Delta A - \frac{1}{\omega C_{rs}} - \frac{1}{\omega C_{rp}}, \tag{9}$$

where

$$\Delta = \frac{\omega^2 M_{tr}^2 Z_{12}^2 Z_{34}^2}{B^2 R_S^2 + (X_t B - Z_{34}^2 Z_{22})^2}, \tag{10}$$

$$B = Z_{22}Z_{33} - Z_{32}^2, \ X_t = -\frac{1}{\omega C_{ts}} - \frac{1}{\omega C_{tp}}, \tag{11}$$

$$A = \frac{(Z_{22}Z_{34}^2 - X_t\omega^2 M_{tr}^2)Z_{34}^2 - Z_{33}B(X_t^2 + R_S^2)}{\omega^2 M_{tr}^2 Z_{34}^2}. \tag{12}$$

The impedance $Z_{out}$ can be seen at the output of the two-port network, which is complex conjugate form of load impedance $Z_L$. This impedance is used to derive the reflection coefficient ($\Gamma$), which can be seen from the load side

$$\Gamma = (Z_{out} - Z_0)/(Z_{out} + Z_0) \tag{13}$$

where $Z_0$ is reference impedance equal to $R_S$.

Equation (13) is used to draw $\Gamma$ in the Smith chart to visualize graphically and estimate the values of load impedance that can be perfectly matched. Furthermore, a derivation of the reflection coefficient expression for the inductive matching is discussed in next section, and numerical results are given in Section 3.

## 2.2. Inductive Matching Network

Inductive coupling is another method widely exploited to match the input impedance for different distances between resonators or load variation. This system uses additional magnetically coupled

coils at the transmitter or receiver, or both, to enhance the PTE. These coils do not need as high Q as the resonator coils. The most popular one is a four-coils system with source coil $L_S$, high-Q transmitter coil $L_t$, high-Q receiver coil $L_r$, and load coil $L_L$ [4,15]. The matching can be controlled by varying couplings between the source/load coils and high-Q coils—$k_S$, $k_L$. The equivalent lossless model of such a system is shown in Figure 4. The high-Q coils are connected to series external capacitors to form resonators.

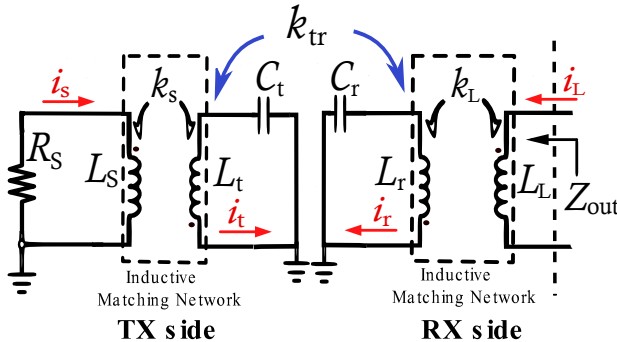

**Figure 4.** Equivalent circuit of WPT system with inductance matching network.

For such a circuit, the same voltage-current relations can be used as in Equation (1), where impedances are

$$Z_{11} = j\omega L_L, \ Z_{12} = -j\omega M_L, \tag{14}$$

$$Z_{21} = -j\omega M_L, Z_{22} = j\omega L_t + \frac{1}{j\omega C_t}, Z_{23} = j\omega M_{tr}, \tag{15}$$

$$Z_{32} = j\omega M_{tr}, Z_{33} = j\omega L_r + \frac{1}{j\omega C_r}, Z_{34} = -j\omega M_S, \tag{16}$$

$$Z_{43} = -j\omega M_S, \ Z_{44} = R_S + j\omega L_S, \tag{17}$$

and by neglecting cross-coupling:

$$Z_{13} = Z_{14} = Z_{24} = Z_{31} = Z_{41} = Z_{42} = 0. \tag{18}$$

Here, $M_{S/L}$ is a mutual inductance between source/load coil ($L_{S/L}$) and high-Q coil ($L_{t/r}$)

$$M_{S/L} = k_{S/L}\sqrt{L_{S/L}L_{t/r}}, \ 0 \le k_{S/L} \le 1. \tag{19}$$

and $M_{tr}$ is the mutual inductance between resonator coils

$$M_{tr} = k_{tr}\sqrt{L_t L_r}, \ 0 \le k_{tr} \le 1. \tag{20}$$

If we assume that the two resonators have same resonance frequency $\omega = 1/\sqrt{L_{t/r}C_{t/r}}$, then

$$Z_{22} = Z_{33} = 0. \tag{21}$$

From impedance Equations (14)–(18) the real and imaginary parts of output impedance $Z_{out}$ become

$$Re\{Z_{out}\} = \Delta R_S, \ Im\{Z_{out}\} = \omega(L_L - \Delta L_S), \tag{22}$$

where

$$\Delta = \frac{\omega^2 M_S^2 M_L^2}{M_{tr}^2 (R_S^2 + \omega^2 L_S^2)}. \tag{23}$$

Furthermore, Equations (13), (22) and (23) are used to calculate the reflection coefficient in the following section. These results conclude the theoretical analysis that is required to compare the two techniques.

## 3. Matchable Regions of Lossless Model

This section presents graphs of reflection coefficients in the Smith Chart, based on the equations derived in the previous section for both capacitive and inductive matching. These graphs help us to estimate the matchable loads and to select the proper matching network at the transmitter and receiver side. In the capacitive method the input impedance is controlled via capacitances $C_{ts}$, $C_{tp}$, $C_{rs}$, $C_{rp}$, whereas in the inductive matching it is controlled by coupling coefficients $k_S$, $k_L$ between the source/load coils and high-Q coils.

In this section, the parasitic components of the system are intentionally excluded to have a clear and ideal case comparison between these two matching techniques. Reflection coefficients were examined for three specific applications, and parameter values used for the cases given in Table 1. The presented cases have been chosen so that they cover a wide range of WPT applications with different specifications for power level and operating frequency [20–22].

**Table 1.** Applications and parameters.

| Application | Operation Frequency | $L_t = L_r$ | $C_t = C_r$ | $k_{tr}$ | References |
|---|---|---|---|---|---|
| Case A—Car charging | 85 kHz | 60 µH | 58.4 nF | 0.01 | [20] |
| Case B—Tablet charging | 6.78 MHz | 6 µH | 91.8 pF | 0.01 | [21] |
| Case C—High Frequency | 100 MHz | 2.5 µH | 1.01 pF | 0.01 | [22] |

Figures 5–10 show the realizable reflection coefficient values in the Smith Chart. The matchable regions are indicated by bold black borders. Each figure consists of three impedance regions, where each region corresponds to different resulting impedances in the circuit: $Z_{tx}$—impedance at the transmitter, $Z_{tr}$—impedance after transmission, $Z_{out}$—impedance at the output. They are obtained by sweeping the impedance matching network parameters over the realistic range of values, which are given in Table 2. Chosen constraint for the inductive IMN is based on an assumption that the driving and load loops have an inductance equal or smaller than the resonator inductances [4]. The bottom limit for the capacitance variance in the capacitive IMN is the lowest value of capacitance in the market, which is approximately 500 fF (ignoring the possibilities of series connection), whereas the upper limit was chosen sufficient for the application choice. As we can see further from the results the upper limit in the inductive method and lower limit in the capacitive method decides the final shape in the Smith chart. The circuits are equivalent models of an inductive WPT, which consists of source resistance $R_S$, matching networks (capacitive or inductive), and lossless coils for transmission and reception.

Case A application in Table 1 is an Electric Vehicle (EV) charging station for transmission of high power. It is designed for low-frequency operation, in our case at 85 kHz frequency, and designed for coils around $L_t = L_r = 60$ µH. Figure 5a shows the impedance at the transmitter. This impedance region agrees well with known results for L-type networks in [23]. It is controlled by varying the capacitances $C_{ts}$ and $C_{tp}$ within the range given in Table 2. Since the L-type capacitive matching network at the transmitter (Tx) cannot match all impedances, a matching network is required for the receiver part as well. Second stage, in Figure 5b, illustrates reflection graph after the resonator coils $L_t$ and $L_r$, which are coupled at $k_{tr} = 0.01$. The inductances change the region into a circle—smaller than Smith chart, which means there is still a limitation in the matchable area. Finally, in Figure 5c, third stage gives impedances that can be matched by the complete network consisting of matching networks at the Tx ($C_{ts}$ and $C_{tp}$) and Rx ($C_{rs}$ and $C_{rp}$) sides. All the capacitances in this example are varied from 0.1 pF to 200 pF (Table 2). The impedance matching network at the Rx side greatly improves the matchable area, which now practically fills the Smith chart. It means that for case A

without losses any load is matchable by the capacitive matching method at $k_{tr} = 0.01$, but it does not mean that this still holds for other coupling coefficient values.

The result of following a similar procedure for case A with the inductive impedance matching network is shown in Figure 6. In this case, matching works by adjusting source/load inductances $L_S/L_L$ and coupling coefficients between these and resonator coils ($k_S$, $k_L$). Inductance and coupling constant ranges are given in Table 2. In Figure 6a, we can notice that the inductive method gives an extremely limited region of impedances that can be matched, hence, matching at the receiver becomes crucial. One thing that can be noticed from Figure 6b impedance range of $Z_{tr}$ after the coupling ($k_{tr} = 0.01$) between resonators $L_t$, $L_r$ is even smaller, and it shows that three-coil system is not suitable for applications where load values are diverse. However, the resulting matchable impedance of the complete system with four coils, where impedance can be matched at both sides (Tx and Rx), fills around 80% of the Smith chart and is shown in Figure 6c. According to the figure, for case A without losses there is roughly 20% of the impedances that cannot be perfectly matched by the inductive approach.

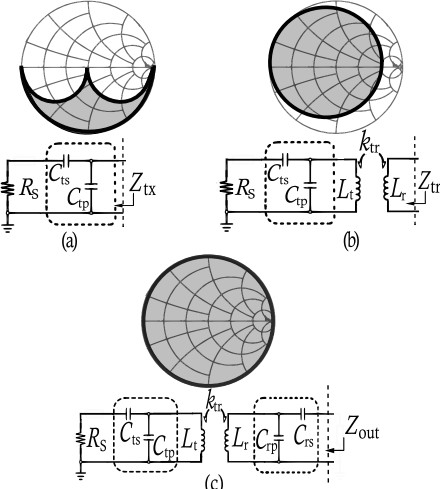

**Figure 5.** Reflection coefficient graphs show impedances that can be obtained by capacitive matching networks placed into WPT system in case A: (**a**) $Z_{tx}$—impedance at the transmitter, (**b**) $Z_{tr}$—impedance after transmission, (**c**) $Z_{out}$—impedance at the output.

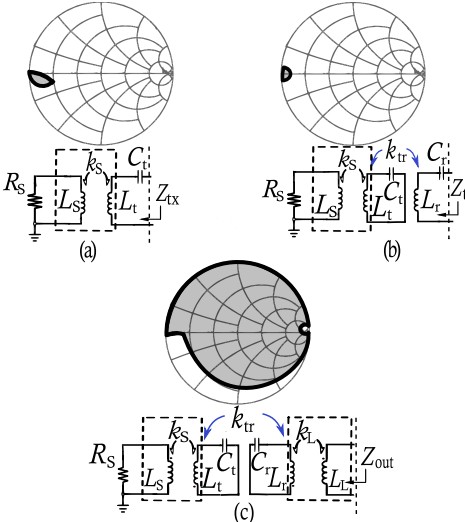

**Figure 6.** Reflection coefficient graphs show impedances that can be obtained by inductive coupling in four coiled WPT system in case A: (**a**) $Z_{tx}$—impedance at the transmitter, (**b**) $Z_{tr}$—impedance after transmission, (**c**) $Z_{out}$—impedance at the output.

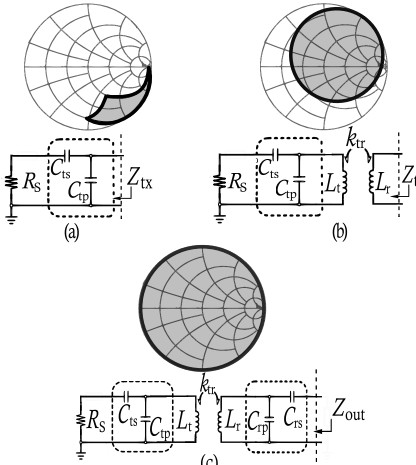

**Figure 7.** Reflection coefficient graphs show impedances that can be obtained by capacitive matching networks placed into WPT system in case B: (**a**) $Z_{tx}$—impedance at the transmitter, (**b**) $Z_{tr}$—impedance after transmission, (**c**) $Z_{out}$—impedance at the output.

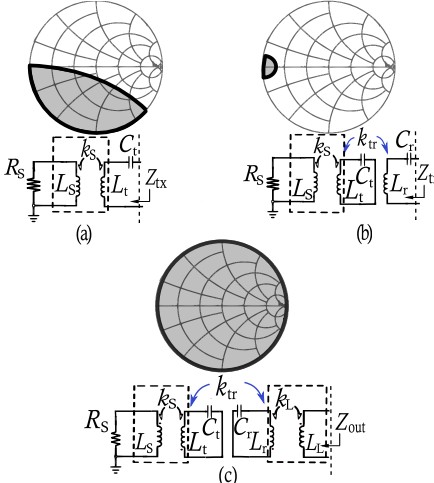

**Figure 8.** Reflection coefficient graphs show impedances that can be obtained by inductive coupling in four coiled WPT system in case B: (**a**) $Z_{tx}$—impedance at the transmitter, (**b**) $Z_{tr}$—impedance after transmission, (**c**) $Z_{out}$—impedance at the output.

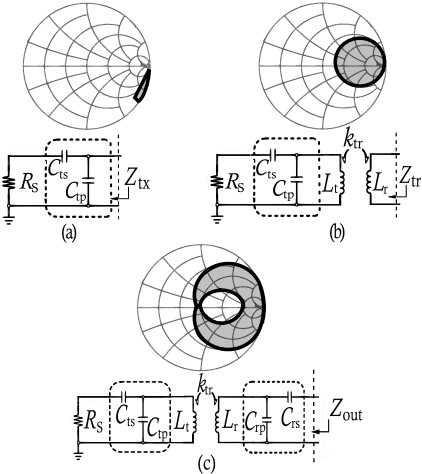

**Figure 9.** Reflection coefficient graphs show impedances that can be obtained by capacitive matching networks placed into WPT system in case C: (**a**) $Z_{tx}$—impedance at the transmitter, (**b**) $Z_{tr}$—impedance after transmission, (**c**) $Z_{out}$—impedance at the output.

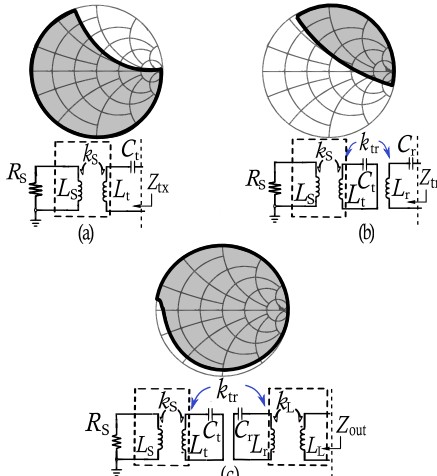

**Figure 10.** Reflection coefficient graphs show impedances that can be obtained by inductive coupling in four coiled WPT system in case C: (**a**) $Z_{tx}$—impedance at the transmitter, (**b**) $Z_{tr}$—impedance after transmission, (**c**) $Z_{out}$—impedance at the output.

**Table 2.** Matching network parameters.

| Application | $L_S$, $L_L$ | $X_S$, $X_L$ | $k_S$, $k_L$ | $C_{ts/rs}$, $C_{tp/rp}$ | $X_{ts/rs}$, $X_{ts/rs}$ |
|---|---|---|---|---|---|
| Case A | 0.1–60 µH | 8.5 mΩ–32.0 Ω | 0.001–1 | 0.1–200 nF | 187 kΩ–9 Ω |
| Case B | 0.1–6 µH | 0.7 Ω–40.7 Ω | 0.001–1 | 0.5–200 pF | 47 kΩ–117 Ω |
| Case C | 0.1–2.5 µH | 10 Ω–1.6 kΩ | 0.001–1 | 0.5–2 pF | 3.2 kΩ–796 Ω |

Case B is a low power and high-frequency system, a mobile phone charging device from Airfuel Alliance, which has standard parameters as 6.78 MHz operational frequency and around $L_t = L_r = 6$ µH coils for Tx and Rx [21]. Results are given in Figures 7 and 8. Here, the inductive method again shows low performance for a three-coil system since the matchable area fills only about 10% of the Smith chart. However, both methods are able to match all loads with the complete network.

Finally, we consider application case C, which is a high-frequency device with $L_t = L_r = 2.5$ µH for Rx and Tx coils. The inductive matching network has larger matchable region than the capacitive method, see Figures 9 and 10. In a complete network, around 90% of the load impedances are matchable by the inductive method, whereas the capacitive approach can match only around 30% of the loads. One thing to note is that the matchable region for the capacitive network has a hole, which appears because of the minimum constraint 0.5 pF in the parameter range. Consequently, the capacitive matching network with these constraints is less versatile in this type of application.

## 4. Performance of Lossy System

In this section, system performance is examined in the presence of loss. Therefore, the lossless inductors $L_t$ and $L_r$ in the previous circuits is replaced by a model of a non-ideal inductor shown in Figure 11. The model consists of an ideal inductor with a series parasitic resistance and a parallel capacitance. We only consider the effect of the parasitic resistance since the parasitic capacitance can be taken care of by compensation circuits. The parasitic resistance is a combination of ohmic and radiative losses of the coil. Other circuit parameters are kept the same as in the previous section for calculation of reflection coefficients. The realizable reflection coefficients over the chosen range of parameters are shown in Figures 12–14.

The inductances of the coils in case A are larger than for the other two cases, so the parasitic resistance $R = 1$ Ω of each coil is higher than for case B and comparable to case C where the skin effect matters. R is estimated assuming a copper coil made from a 30-m long wire of diameter 0.8 mm [20]. The result for case A is shown in Figure 12, where Figure 12a presents realizable reflection coefficients

for capacitive IMN of Figures 3 and 12b presents the results for inductive IMN of Figure 4. Solid lines show borders of matchable regions for lossles networks determined in the previous section, whereas dashed lines bound matchable regions for lossy networks.

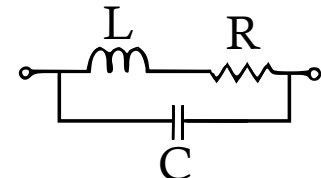

**Figure 11.** Model of non-ideal inductor.

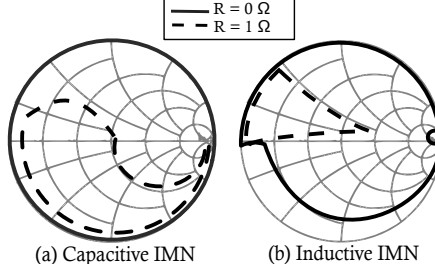

(a) Capacitive IMN   (b) Inductive IMN

**Figure 12.** Matchable regions in lossless and lossy model for case A.

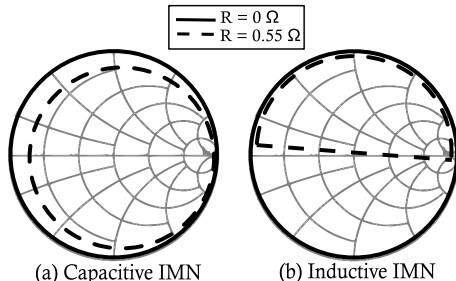

(a) Capacitive IMN   (b) Inductive IMN

**Figure 13.** Matchable regions in lossless and lossy model for case B.

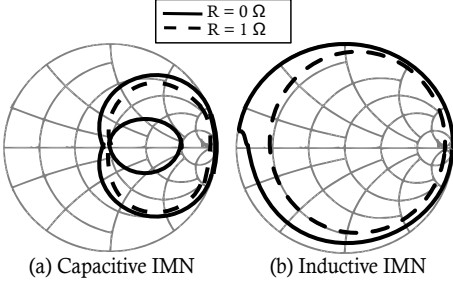

(a) Capacitive IMN   (b) Inductive IMN

**Figure 14.** Matchable regions in lossless and lossy model for case C.

The matchable area has been dramatically reduced from 100% to approximately 40% of the Smith chart for capacitive and from 80% to 20% for the inductive technique. It shows that range of matchable impedances is sensitive to loss for both methods and that the inductive method is slightly sensitive to the loss than the capacitive method.

The comparison of matchable regions, after the introduction of parasitic resistance $R = 0.55\ \Omega$ ([21]) in the coils for case B, is shown in Figure 13. The reflection coefficient graphs of capacitive and inductive methods are shown in Figure 13a,b, respectively. The change in the matchable region is again more dramatic for the inductive method since it has about 55% reduction from the ideal case compared to the capacitive method's 20%.

Matchable regions for case C are given in Figure 14. Since the resonator coils operate at the highest frequency in the comparison, we considered skin effect as well. Therefore, ohmic loss at the coils is

chosen as a combination of parasitic resistance and skin effect loss, which is $R = 1\ \Omega$. R is estimated assuming a copper coil made from a 75-cm long wire of diameter 0.8 mm [22]. As a peculiarity in Figure 14a, we note that the matchable region for the lossless capacitive network has a hole in it and that this vanishes when losses are introduced. The overall numbers are 2% reduction in area for the capacitive and 15% for the inductive approach. While the inductive method in this case still has a matchable region more sensitive to loss than capacitive IMN, the inductive method covers a larger area of matchable impedances both with and without loss.

As mentioned before in Section 2, we considered regions of matchable impedances. It can also be interesting to see how well the network performs for any given load impedance. Here we consider this question by calculating the delivered power to load impedances sampled form the entire Smith chart. It can be obtained in several ways, but we used SPICE AC analysis here. We focused on cases A and B because their matchable regions are most affected by the introduction of ohmic loss. The results are shown in Figures 15 and 16 and demonstrate that capacitive and inductive IMNs can be comparable in matching various load impedances, when the networks are optimized for each particular impedance. Device in case A has more power loss than in case B. Overall, power level is distributed between $-4.2$ dBm and $-4.8$ dBm for case A, and $-3.5$ dBm and $-4$ dBm for case B.

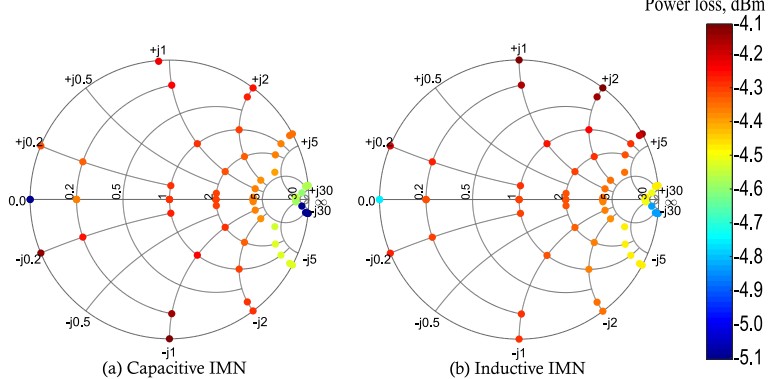

**Figure 15.** Comparison of optimal power delivered to the various loads for case A.

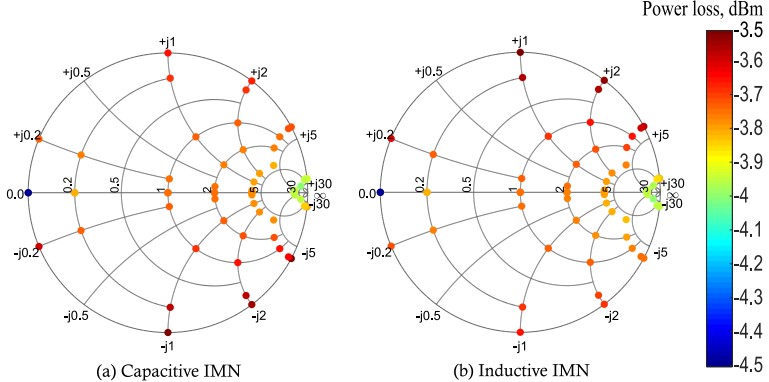

**Figure 16.** Comparison of optimal power delivered to the various loads for case B.

## 5. Discussion

Section 3 presented three impedance areas obtained for different stages of ideal circuits (Figures 5–10), which is summarized in Figure 17. In this figure, each trace represents specific application (case A, B, C) and matching network. The impedances at the different stage of circuits, in Figures 5–10, are given in *x*-axis, whereas *y*-axis shows a share of Smith chart in percentage. All the areas have been estimated from the graphs. According to the figure, $Z_{out}$ fills a larger area than $Z_{tr}$ for all cases, which shows that the impedance matching network at the Rx side is crucial in obtaining large tunable area of load impedance. There is a degradation from 100 to 30% in the area of output

impedance $Z_{out}$ by the capacitive method from case A to C. An improvement from 10 to 75% in the area of impedance at the transmitter side $Z_{tx}$ can be seen for the inductive technique from case A to C. This is due to difference in the operation frequency of the specific applications since for capacitive IMN $X_C \propto 1/f$, while for inductive IMN $X_L \propto f$. Therefore, the region of inductive method shrinks in case C (85 kHz—low frequency) and widens in case A (100 MHz—high frequency), and it is vice versa for capacitive matching. At low operation frequencies, the available range of the capacitance values is enough to match most of the load impedances, whereas at high frequency, the matching is limited by the smallest capacitance value in the parameter range, which is 0.5 pF. In the inductive approach, the ranges of load and source inductance values are the same as for the resonator coils. For case C, covering most of the Smith chart can be achieved with a reasonable inductance range, whereas for case A, the inductance range is not large enough to match all the load impedances.

The limitation discussed above also gives a better understanding of the parasitic effects in the lossy WPT systems. For capacitive method, it is crucial to avoid parasitic capacitances of resonator coils at high frequencies since they are comparable with 0.5 pF, whereas for low frequency applications they can be ignored.

The matchable areas of the lossless model and the circuit with parasitics are compared in Figure 18. Here, blue/yellow colors correspond to matching network type and solid/hatched patterns of charts represent ideal and lossy scenarios of examination. The figure shows that the circuit with inductive IMN is more sensitive to the loss than capacitive IMN. In case A, the reductions in matchable areas of inductive and capacitive IMNs are comparable. In case B and case C, the reduction is higher for inductive IMN than for capacitive IMN. Case C is least affected by the parasitics due to high operation frequency. It should be noted that a larger parasitic resistance shrinks the matchable region, whereas a smaller parasitic resistance increases it closer to the ideal case.

It is clear that matching networks provide quite different areas of perfectly matchable impedances. However, the comparison in Figures 15 and 16 showed that the methods have comparable power transfer when the networks are optimized for each load.

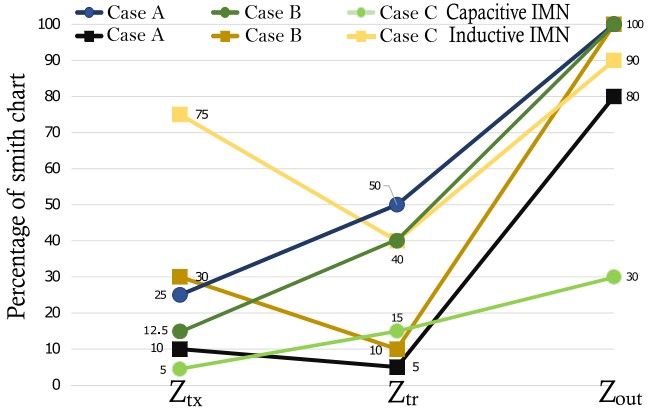

**Figure 17.** Matchable area comparison for different stages of the circuit in the ideal case.

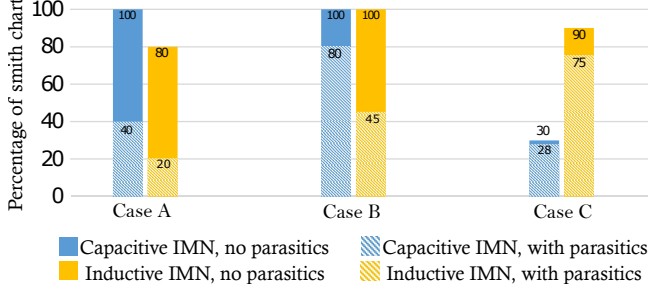

**Figure 18.** Area of matchable loads before and after introducing the parasitics.

## 6. Conclusions

In this paper, we compare areas of matchable loads for capacitive and inductive impedance matching networks (IMNs), which are the common matching techniques in the magnetic resonant wireless power transfer system. Graphical visualization of the impedances by Smith Chart is used for effortless comparison of the IMNs performances. An analytic expression for effective output impedance is derived and used to display the conjugate-image impedance of the load. Without any limitations in the parameter values and frequency range, it is always possible to match any load, i.e., any point in the Smith chart. Therefore, three different applications were considered with constraints: case A—car charging operating at 85 kHz, case B—mobile phone charging at 6.78 MHz, and case C— a high-frequency charging device at 100 MHz.

For the lossless system, the capacitive circuit's matchable area fills the Smith chart for case A, whereas around 20% of the chart that cannot be matched by the inductive IMN. For case B both methods could match any load impedance. On the other hand, the inductive IMN has shown about 60% larger area than the capacitive IMN for case C. The matching network at the receiver improves the area in all cases.

Finally, the impact of parasitic resistance of the resonator coils to the matchable area has been examined. In case A, the reduction of matchable area is 60 percentage points for both methods. The matchable area by the inductive IMN is more sensitive to the parasitic resistance than capacitive IMN for cases B and C. However, simulation of power transfer has shown that both matching networks can be equally effective in matching different load values.

**Author Contributions:** All authors contributed to problem formulation, conceptualization and choice of methods. Y.Z. performed all calculations, simulations and original draft preparation. All authors contributed to interpretation of results, and to editing into the final manuscript. All authors have read and agreed to the published version of the manuscript.

**Funding:** Y.Z. benefits from a PhD scholarship from University of South-Eastern Norway funded by the Norwegian Ministry of Education and Research. Article processing charge (APC) was funded by the University of South-Eastern Norway.

**Acknowledgments:** The Research Council of Norway is acknowledged for the support to the Norwegian Ph.D. Network on Nanotechnology for Microsystems, Nano-Network (221860/F40).

**Conflicts of Interest:** The authors declare no conflict of interest.

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
