# Peer review of "Analysis of Fundamental Differences between Capacitive and Inductive Impedance Matching for Inductive Wireless Power Transfer"

_electronics, doi:10.3390/electronics9030476_

Round 1

Reviewer 1 Report

In this paper, the basic impedance matching techniques using capacitors and inductors respectively are evaluated and compared. Three different applications are chosen as the use cases to parameterize the impedance matching networks, and the evaluations are performed in both ideal and non-ideal circumstances. Generally, the comparison provided by this paper is fundamental and not particularly new, but the results are supported by a very clear demonstration, which makes them convincible. A detailed discussion is given and the overall presentation is well done. In my opinion, only a few issues need to be addressed, as follows:

  1. In Section 2.1, a capacitive matching network is demonstrated and reflection coefficient is derived. The authors use serial-parallel compensations at both the Tx and Rx sides. While the connections may affect the power transfer efficiency, would the results become different if a different compensation structure, for instance parallel-serial compensation at both sides, is used? This needs to be explained.

  1. In Section 4, the performances for the networks with parasitic resistance and capacitance are evaluated. In the three cases, the parasitic resistance is given as 1Ω, 0.55Ω and 1Ω. How are these values are determined? Are they simply based on some assumption or given by a theoretical calculation? This is missing in this paper. In my understanding, the parasitic resistance would definitely affect the matchable regions and the power transfer efficiency, so this is quite crucial for the results and the conclusions. A detailed explanation has to be included.

Author Response

We appreciate all the efforts from the reviewers and editors involved in the review process, since it helps us to improve the paper further. According to reviewers’ comments, the changes has been made and highlighted in red in the paper.

Review 1

Comments to authors:

In this paper, the basic impedance matching techniques using capacitors and inductors respectively are evaluated and compared. Three different applications are chosen as the use cases to parameterize the impedance matching networks, and the evaluations are performed in both ideal and non-ideal circumstances. Generally, the comparison provided by this paper is fundamental and not particularly new, but the results are supported by a very clear demonstration, which makes them convincible. A detailed discussion is given and the overall presentation is well done. In my opinion, only a few issues need to be addressed, as follows:

  1. In Section 2.1, a capacitive matching network is demonstrated and reflection coefficient is derived. The authors use serial-parallel compensations at both the Tx and Rx sides. While the connections may affect the power transfer efficiency, would the results become different if a different compensation structure, for instance parallel-serial compensation at both sides, is used? This needs to be explained.

There are many capacitive compensation circuitries and we chose the parallel-series compensation according to the reference [9] in order to compare it to the inductive method. This network already has sufficient design degrees of freedom to match perfectly if there is no loss and no restriction on parameter ranges, hence it is sufficient to give insight into the effect of these limitations. If the compensation circuitry was parallel-series, the matchable regions would be different and our method can be used to showcase that as well.  A comment has been added to the introduction.

  1. In Section 4, the performances for the networks with parasitic resistance and capacitance are evaluated. In the three cases, the parasitic resistance is given as 1Ω, 0.55Ω and 1Ω. How are these values are determined? Are they simply based on some assumption or given by a theoretical calculation? This is missing in this paper. In my understanding, the parasitic resistance would definitely affect the matchable regions and the power transfer efficiency, so this is quite crucial for the results and the conclusions. A detailed explanation has to be included.

R was estimated through calculation for each case.

For Case A, R was calculated with parameters: L= 30m, d=0.8 mm, f=85 kHz [21].

For case B it is taken from the reference [22].

For case C it was estimated for the copper coil: L=75 cm, d=0.8 mm, f=100 MHz [23].

Determination of parasitic resistance to each case is added in the section 3 and highlighted red.

Reviewer 2 Report

In this paper, the authors have represented a new comparison method by determining the matchable impedances of the selected networks. The comparison is visualized as regions of a Smith chart. The study is very interesting, however, several minor concerns in the current stage of manuscript need to be addressed before consideration of acceptance, as follows:

To make the manuscript self-contained and more readable, make sure all the variables have been fully specified after each equation.

Please correct the font of all the variables to make it consistent in both the context and figures.

Please specify the variable in Figure 1 and 2.

In section 3, please add more motivations for comparing the presented cases – A, B and C. Does any other case setup exist?

What is the varying trend of the matchable region for an extended range of resistance R? It is suggested to add more discussions on the effect of R with more extensive range.

Please add the units for color bar in Figure 15. Moreover, to make it comparable, it’s recommended to use the same color bar for Figure 15 and 16.

Carefully recheck residual errors.

As such the subject matter is of interest to those in the field of inductive wireless power transmission, more specifications on other power transmission devices could be added. For example, when mention ‘transfer of high power and low-frequency operation’, two research works using capacitive nano-generators for the power transmission could be highlighted – A self-powered radio frequency (RF) transmission system based on the combination of triboelectric nanogenerator (TENG) and piezoelectric element for disaster rescue/relief (radio frequency transmission); A model for the triboelectric nanogenerator with inductive load and its energy boost potential (using the resonance between the capacitive nano-generators and inductive load to boost the transfer efficiency)

Author Response

We appreciate all the efforts from the reviewers and editors involved in the review process, since it helps us to improve the paper further. According to reviewers’ comments, the changes has been made and highlighted in red in the paper.

Reviewer 2

Comments:

In this paper, the authors have represented a new comparison method by determining the matchable impedances of the selected networks. The comparison is visualized as regions of a Smith chart. The study is very interesting, however, several minor concerns in the current stage of manuscript need to be addressed before consideration of acceptance, as follows:

To make the manuscript self-contained and more readable, make sure all the variables have been fully specified after each equation.

The comment is noted, and changes are made accordingly.

Please correct the font of all the variables to make it consistent in both the context and figures.

Figure font changes are made.

Please specify the variable in Figure 1 and 2.

The variables in Fig 1 and 2 are now specified in the figures and referred to  in the text.

In section 3, please add more motivations for comparing the presented cases – A, B and C. Does any other case setup exist?

Wireless power transfer has many applications at different operation frequencies and power levels. We chose these cases, so we cover a wide range of scale applications. This motivation on the choices are added to the Section 3, and highlighted red.

What is the varying trend of the matchable region for an extended range of resistance R? It is suggested to add more discussions on the effect of R with more extensive range.

Certainly, an extended range of R will influence the matchable regions: smaller R will give us closer to the ideal case result, whereas increment of parasitic resistance will shrink matchable region further.  This explanation has been added to section 5. 

Please add the units for color bar in Figure 15. Moreover, to make it comparable, it’s recommended to use the same color bar for Figure 15 and 16.

Figures 15 and 16 have been arranged differently with individual colour-bars.  Each colour-bar has separate unit.

Carefully recheck residual errors.

We have carefully checked for errors.

As such the subject matter is of interest to those in the field of inductive wireless power transmission, more specifications on other power transmission devices could be added. For example, when mention ‘transfer of high power and low-frequency operation’, two research works using capacitive nano-generators for the power transmission could be highlighted – A self-powered radio frequency (RF) transmission system based on the combination of triboelectric nanogenerator (TENG) and piezoelectric element for disaster rescue/relief (radio frequency transmission); A model for the triboelectric nanogenerator with inductive load and its energy boost potential (using the resonance between the capacitive nano-generators and inductive load to boost the transfer efficiency)

These are interesting papers, but we do not think they are sufficiently relevant to the scope of our paper.